# Morula Tree: From Fruit to Wine through Spontaneous Fermentation and the Potential of Deriving Other Value-Added Products

**Lesetja Moraba Legodi** \*, **Maleho Annastasia Lekganyane** and **Kgabo L. Maureen Moganedi**

Department of Biochemistry, Microbiology and Biotechnology, University of Limpopo, Private Bag X 1106, Sovenga 0727, South Africa

\* Correspondence: lesetja.legodi@ul.ac.za

**Abstract:** *Sclerocarya birrea* (Morula tree) is one of the indigenous trees bearing wild fruits with various applications in the African communities. Wine is a globally known beverage usually made from grapes; however, recently, other fruits, including wild fruits with a considerable amount of sugars, can be used for making wines. The marula fruit wine is also important in many communities for cultural activities and can be enjoyed by people of varying age groups depending on the age of the product. In recent years, there has been growing interest in shifting from traditional marula winemaking to developing technologies for the marula winemaking process and commercialisation. The process of marula winemaking is similar to the production of grape wines, which entails collection, selection and washing of the fruits; extraction of the juice and mashing; formation and removal of the scum; and ultimately spontaneous fermentation of the resulting juice. The new process in marula winemaking would take into consideration the use of starter cultures as either monoculture or mixed cultures developed from the native marula fruit microbiota and the pasteurisation of the juice. The main challenge or difficulty with marula is the extraction of sugar and other soluble solids from the pulp more than it is for the grapes. The other challenge confronting the sustainability of marula wine is the seasonality of the fruit and poor juice yield. It is therefore imperative to develop strategies to increase the juice yield without affecting the quality, to preserve the marula fruits to ensure the year-round presence of marula fruit wine in the markets and, consequently, to improve the income generation capacity of the households dependent on the product. In addition to achieving a high juice yield, it is imperative to ensure consistent quality wine products. This review gives an overview of the *S. birrea* subsp. *caffra* and the biochemical components of the fruits or juice. It also highlights the use of marula fruits for wine production in African communities. The potential economic sustainability of the marula fruit wine is explored, particularly in southern Africa, where the marula tree (Morula) is abundant and the marula fruit wine is popularly produced. The review also examines the opportunities, challenges and future prospects of the marula fruit wine.

**Keywords:** *Sclerocarya birrea*; marula fruits; marula fruit wine; marula juice; wine fermentation

## 1. Introduction

*Sclerocarya birrea* subsp. *caffra* (also known as morula in Northern Sotho) is an indigenous tree of southern Africa. The tree belongs to the Anacardiaceae family and class magnoliopsida. The tree is drought resistant and available in Namibia, Botswana, Zambia, Swaziland, Zimbabwe and South Africa. In South Africa, the tree grows abundantly in different regions such as northern Kwa-Zulu Natal, Mpumalanga and Limpopo Province [1]. *S. birrea* subsp. *caffra* is widely used by rural communities for various purposes to meet the needs of their livelihood. Every part of the tree including the fruits, branches, stem (bark) and roots is essential for cultural, social and economic activities within African communities [1–3]. The fruits of *S. birrea* subsp. *caffra* are called marula.

Anatomically, *S. birrea* subsp. *caffra* is a deciduous tree and can grow tall from 7 to 17 m in height [4]. Morula tree has a single round stem that is covered by grey bark with long narrow cracks, which forms branches at 1 m or more above ground (Figure 1). The leaves are divided into 10–18 pairs of leaflets, each about 8–60 mm long and are oval-shaped with smooth edges. The leaflets are dark-green above and paler or bluish-green below [5,6]. The marula fruit size is variable but roughly plum-sized. The fruits abscise before ripening, when the skin colour is still green and the fruit is still firm. The ripe fruits have a thick yellow skin and translucent whitish flesh. The outer skin odour has been described as strong apple-like, and marula fruit pulp has a flavour that is likened to a mixture of litchi, apple, guava and pineapple. It has also been noted that the fruit size and flavour vary from one tree to another tree [2].

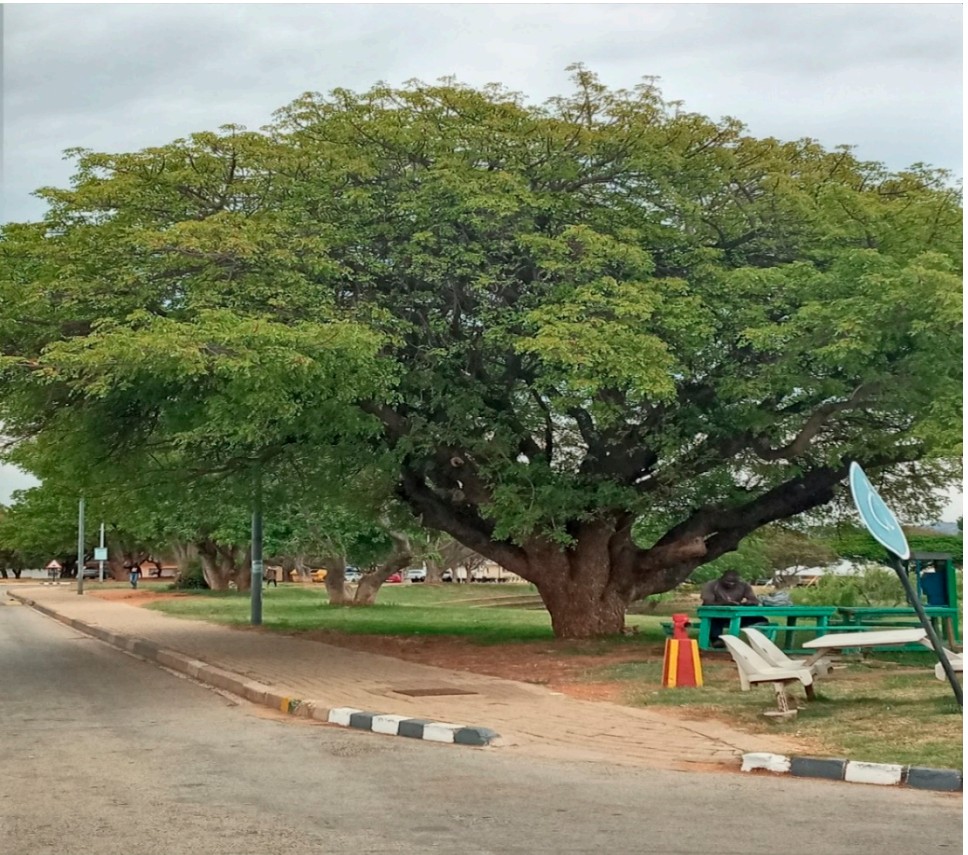

**Figure 1.** Morula tree at the University of Limpopo, South Africa (Source: image taken in October 2021 by Author).

In Limpopo province, the Pedi (Northern Sotho) people recognise three varieties of morula trees and their fruits based on the aroma and flavour, namely, *morula o mobose*, *morula wa go baba* and *morula wa go nkga*.

(1)   *Morula o mobose* is described as the tree which bears sweet, palatable fruits.
(2)   *Morula wa go baba* is the tree bearing sour and undesirable fruits.
(3)   *Morula wa go nkga* bears fruits that are disliked due to their "objectionable odour".

These examples of fruit type differentiation in Pedi terminology reflect the social importance of this species, which typify the taxonomy of *S. birrea* subsp. *caffra* in southern Africa [2]. The sweet fruit-bearing tree should be selected and protected for fruit production in order to meet the demands for the marula fruit wine markets and the new markets for fruit juices. The marula fruit wine is also called the traditional marula beer or just marula beer, and for the subsequent sections, it is referred to as marula fruit wine. The good cultivar selection would increase consistency in the product and subsequently increase productivity and provide an incentive for the farmers, including communities [7].

The exploitation of *S. birrea* subsp. *caffra* and its products form part of South Africa's expansion programme on indigenous knowledge systems (IKS). The tree has been identified as a key driver in the development of rural community businesses for various products and traditional medicine. This paper focuses mainly on the processing of the marula fruit for marula fruit wine and various other useful products, as illustrated in Figure 2. The end products of fruit processing are kernels, juice and peels. The quantity of the kernels produced depends on the geographic location and the cultivar. For instance, the mean kernel mass (0.42 g) was significantly greater in fruit-nuts harvested from the farmer's field than in the fruit-nut obtained from communal land (0.30 g) and 0.32 g kernel from natural woodland fruit nuts [8]. Large quantities of marula fruits are collected each season, primarily to eat fresh or used to prepare non-alcoholic juice or alcoholic beverage (fermented juice) and jam. The fruit nut encloses a soft white kernel that is rich in oil and contains 28.4–53.0% proteins [9–14]. The marula seed (i.e., kernels) oil is rich in oleic (67.2–85.2 g/100 g), palmitic (9.65–14.1 g/100 g), myristic (0.33 g/100 g) and stearic acids (5.11–8.84 g/100 g) [5,15,16] and the amino acids are predominantly glutamic acid (10.78% dry matter) and arginine (6.36% dry matter) [5,15]. The kernels are eaten raw, fried or used to extract oil [13,17]. The potential of marula fruit pulp and peel (skin) as substrates for the production of vinegar (acetic acid) through surface/solid-state fermentation and submerged fermentation yielded acetic acid of between 41–57 g/L and 41–54 g/L, respectively. The solid-state fermentation was the preferred method for the production of vinegar because it was non-toxic/non-destructive to the fermenting microorganisms and the conditions favoured optimal metabolic processes [1].

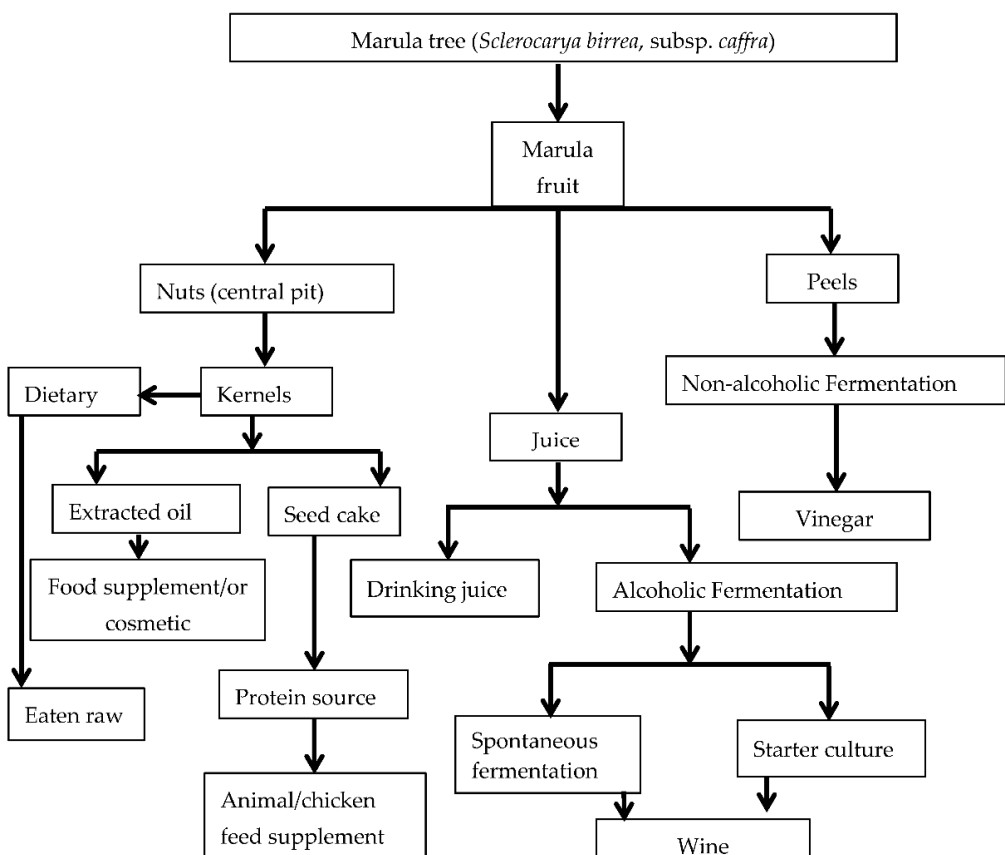

**Figure 2.** Bioprocessing of marula tree for the production of value-added products.

The marula-derived products are increasingly entering the market, either through the local rural communities or by private sector companies with the intent to improve the livelihood of the communities [18,19]. Efforts to commercialise marula products date back to the 1970s. To date, there is one popular alcoholic beverage that has been commercialised

by the Distell SA. The rural communities around the town of Phalaborwa in Limpopo province, South Africa, collect the fruits and deliver them to a centralised location. Only 30% of the collected fruits are taken by Distell SA to make Amarula Cream Liqueur, which has an alcohol content of 17% (*v/v*), while a significant percentage of the fruits are classified as substandard fruits and therefore not utilised and discarded as waste [1].

More emphasis in the subsequent sections will be on marula juice and fruit wine. To date, the marula fruit wine is mainly still produced for cultural and social activities within communities and, to a lesser extent, for sale.

## 2. The Extraction of Marula Fruit Juice

### 2.1. Marula Fruit Juice

The processing techniques for extracting juice from marula fruits differ among villages, regions and countries where the marula tree is dominant [20]. This is a laborious process that usually takes 3–4 h to produce between 20 and 50 L of juice, depending on the juiciness of the fruits and the speed at which the extraction is taking place. Extraction of juice, as outlined by Shackleton [18], is carried out by removing the skin from the fruit using a cow horn or fork. The skin is split open, deftly turned inside out, then separated from the pulp (flesh) and discarded. The fruits are squeezed, and the juice is collected in a clean bucket (concentrate juice). The nuts with some pulp are put in another bucket and covered with water of equal volume as the concentrated juice, where they are pulverised with a wooden spoon to remove excess juice and pulp and then hand squeezed to release residual juice and pulp. The extracted juice is left overnight, where it cleans itself through the formation of a scummy top layer which is commonly scooped out and discarded. Marula fruit juice at this stage has a very short shelf life as a result of enzymes or microbial growth unless it is immediately processed or pre-served [21]. The manual extraction employed for marula fruit juice gives a low yield of juice and other soluble materials from the pulp. The marula fruit juice naturally has low sugar content and higher organic acid content when compared to grapes. The yield of the wine is often enhanced by the addition of water to the juice, which also reduces acidity, while the addition of sugar corrects its deficiency in the juice [22]. There is a need to improve the extraction process of marula juice to make it cost-efficient. A detailed process for spontaneous fermentation and the microbial evolutions and dynamics that take place during the fermentation are elucidated in Section 4.

### 2.2. Biochemical Properties of Marula Juice

The marula fruit juice has a good nutritional value, such as vitamin C, oleic acid and antioxidants [5]. The juice composition is influenced by the marula tree cultivar, soil conditions and the maturity (ripeness) of the fruit [22]. The composition of the marula juice affects not only the fermentation but also the sensory quality of the marula wine. The unfermented marula juice contains sugars in the form of (1.4 g/100 mL of pulp) sucrose, (0.4 g/100 mL) fructose and (0.5 g/100 mL) glucose [12]. The reported sugar level (i.e., total soluble solids) in marula juice varied between 7–14 Brix (°Bx) [22–24]. The main sugar in marula juice is sucrose, with fructose being the least [12], and this was in congruent with the reported order in terms of concentrations (i.e., sucrose > glucose > fructose) by [25]. In other studies, glucose is reported to be the least sugar [20,26,27]. The overall sugar content and juiciness of the marula fruit are affected by the climatic conditions and the amount of rainfall in a particular year [28].

A proximate analysis of marula juice from different studies is shown in Table 1 [26,29] in comparison to the marula fruit [30]. Marula fruit contained a high amount of sodium, potassium and calcium [30], whereas, in the juice, calcium remained low while magnesium content was high [26,29]. The analysis of the nutritional value of marula suggests that the consumption of marula juice should be encouraged amongst people of all ages because of the high natural antioxidant content and minerals.

**Table 1.** Proximate analysis of marula fruit and juices.

| Components | Marula Fruit [30] | Marula Juice [26] | Marula Juice [29] |
|---|---|---|---|
| **Macronutrients** | Percentage (%) | g/100 mL | g/100 mL |
| Moisture | 81–91.7 | * ND | * ND |
| Ash | 10.4–15.4 | 1.01 ± 0.03 | 5.05 ± 0.61 |
| Protein | 9.08–9.93 | 0.31 ± 0.04 | 3.31 ± 0.10 |
| Crude lipid | 0.9–1.2 | * ND | 1.30 ± 0.15 |
| Crude fibre | 6.46–7.03 | 0.70 ± 0.12 | * ND |
| Total soluble solid (TSS) | * ND | * ND | 12.32 ± 1.02 |
| Carbohydrates | 55.97–61.28 | * ND | 90.35 ± 0.77 |
| Sucrose | * ND | 6.2 ± 0.8 | 0.76 ± 0.21 |
| Glucose | * ND | 0.5 ± 0.3 | 0.21 ± 0.01 |
| Fructose | * ND | 0.6 ± 0.4 | * ND |
| **Micronutrients** | mg/100 g | mg/100 mL | mg/100 mL |
| Sodium | 36.64–41.01 | 10 ± 2.00 | 14.88 ± 6.00 |
| Potassium | 69.54–250 | 328 ± 11 | 44.54 ± 0.41 |
| Calcium | 317.33–842 | 40 ± 6.00 | 51.73 ± 6.00 |
| Phosphorus | 0.23–0.37 | * ND | 0.18 ± 0.02 |
| Magnesium | 9.27–12.93 | 44 ± 4.00 | 24.53 ± 2.06 |
| Nickel | 0.43–5.8 | * ND | 0.21 ± 0.10 |
| Manganese | 0.67–1.43 | 0.05 ± 0.01 | 6.60 ± 4.10 |
| Copper | 0.31–1.2 | * ND | 1.07 ± 0.10 |
| Zinc | 0.41–1.22 | 0.19 ± 0.02 | 2.96 ± 1.0 |
| Iron | * ND | 0.071 | 0.883 ± 0.15 |

* ND refers to "not determined".

A previous study [23] compared the proximate composition and nutritional value of marula fruit with other tropical and indigenous fruits. The authors inferred that the nutrient content was greatly influenced by the processing methods, geographic place, soil, climate and analytical methods used. As shown in Table 2, the nutrient and mineral content of marula fruit and the juice are influenced by the place of origin, soil type, climate and the time that lapsed post-harvesting prior to analysis [23,26].

**Table 2.** Comparative analysis of vitamin C content in different fruit juices.

| Citrus Fruits and Other Fruits | Vitamin C Content in Juices (mg/100 g) | References |
|---|---|---|
| Orange (*Citrus sinensis* (L.) Osbeck) | 36–74 | [31] |
| Mandarin varieties<br>Clementines (*Citrus reticulata*, Blanco)<br>Satsuma (*Citrus unshiu* Mark)<br>Tangerine (*Citrus deliciosa* Ten) | 24.2–53.1<br>21.8–47<br>19–54 | [31] |
| Lemon (*Citrus limon* Burm) | 22–61 | [31] |
| Grapefruit (*Citrus paradise* Macf.) | 22.2–78 | [31] |
| Pineapple (*Ananas comosus*) | 68 | [31] |
| Pummelo (*Citrus grandis* Osbeck) | 31.4–47.2 | [31] |
| Marula (*Sclerocarya birrea*) | 62–400 | [20] |
| Apple (*Malcus domestica*) | 7.94–27.3 | [32–34] |
| Papaya (*Carica papaya*) | 45–55.6 | [35] |
| Grape (*Vitis vinifera*. L.) | 0.86–1.36 | [36] |
| Table grape (*V. vinifera*. L.) | 12.76–16.34 | [37] |

Marula juice has vitamin C (ascorbic acid) content ranging from 62 to 400 mg/100 g of fresh juice [20]. This is higher than most fruits such as oranges, grapes, lemons, apples and papaya (Table 2). The variation in vitamin C content of fruits, as reported in different studies, is attributed mainly to the environmental conditions of the place of origin, such as area soil type and climate, fruit ripening stage and the time it takes to do analysis after the harvest [23]. Notwithstanding the fact that the vitamin C content varies between the same fruit as a result of specific cultivar of the tree, there is also the influence of the analytical method used for determining the vitamin C.

In addition, marula fruit has an antioxidant capacity of between 8 and 25 mM ascorbic acid equivalents and total phenolic content ranging from 7.5 to 24 gallic acid equivalent (GAE) in mg/g dry weight [10]. The indigenous fruit contains minute quantities of vitamins such as thiamine, riboflavin, and nicotinic acid and various organic acids such as citric, lactic, malic, tartaric, oxalic, isocitric and acetic acid [38]. Citric acid is the most abundant acid found in the marula fruit; however, the presence of malic and tartaric acid has also been reported [23]. *S. birrea* fruit showed to contain citric acid (8.5 g/100 g), malic acid (1.2 g/100 g), succinic acid (0.1 g/100 g) and tartaric acid in trace amounts in the proximate analysis of five indigenous fruits of Mozambique [12]. In grapes and must, citric acid is present in lower concentration (0.1–1 g/L), whereas tartaric (2–8 g/L) and malic acids (1–7 g/L) constitute major organic acids [39].

### 2.3. Flavour and Aromatic Compounds of the Marula Fruit and Juice

Marula fruit has a unique aroma (smell). The pineapple-like flavour in marula juice is in part due to the complementary volatile compounds such as ethyl acetate, benzaldehyde and linalool [40]. Limited research has been conducted to profile the aromatic compounds of the fruits and the extracted juice. However, few research studies profiled the volatile components from different parts of the fruits. Table 3 shows the 39 components of volatile aroma extracted from the skin with Freon 12 [22,41] and the aroma volatiles obtained by liquid–liquid extraction or head-space analysis in marula juice followed by GC and GC-MS for identification [42].

Table 3 depicts thirty of the volatiles identified in the head-space of intact fruit using solid-phase micro-extraction (SPME) followed by GC-MS, which represented 88.7% of the total composition [42]. The volatile components in marula juice obtained through extraction with Freon 12 accounted for over 95% of the total volatiles, thus from α-ylangene to benzenemethanol. The extraction of the skins with Freon 12 produced the authentic marula flavour [22]. With the head-space analysis of the marula juice, the major aroma compounds were sesquiterpene hydrocarbons and benzyl alcohol [22,41]. The esters and hydrocarbons were the dominant compounds in the fruit. The major compounds included heptadecene (16.1%), benzyl 4-methyl pentanoate (8.8%), benzyl butyrate (6.7%), (Z)-13-octadecenal (6.2%) and cyclo-pentadecane (5.7%). The major alcohol compounds were (Z)-3-decen-1-ol (8.4%) and 6-dodecen-1-ol (3.8%), whereas the major aldehyde was 11-hexadecanal (4.4%). Alcohols such as n-pentanol, 3-methoxy-2-butanol and 2-methyl-1-pentanol were not detected in the fruit [42]. Only three components were identified in marula fruit pulp, namely β-caryophyllene (91.3%), α-humulene (8.3%) and germacrene D (0.1%). The monoterpene hydrocarbons were absent in both the fruit and pulp [42].

It is noteworthy to indicate that the volatile compounds present in the marula skin and the juice differ according to the marula tree cultivars. The limited scientific knowledge of the marula fruit flavour and aroma opens an opportunity to investigate the chemistry-biology that underlies such complex phenomena.

Table 3. Different volatile compounds extracted from intact marula fruit, skin and juice.

| Marula Skin [22] | | Intact Fruit [41] | | | | Juice [42] | |
|---|---|---|---|---|---|---|---|
| n-pentane | Ethyl caproate | Ethyl isovalerate | Benzyl 4-methylpentanoate | Ethyl acetate | | ϒ-amorphene | Tetrahydro-2-*H*-pyran-2-one |
| n-hexane | Benzyl acetate | Ethyl hexanoate | Benzyl tiglate | Ethyl-3-methylbutanoate | | a sesquiterpene hydrocarbon | α-humulene |
| Benzene | | Ethyl octanoate | Hexadecanal | 3-methyl-1-butanol | | ϒ-muurolene | Aromadendrene |
| 2-octene | Glycolic acid | Isoamyl hexanoate | | Pentan-1-ol | | three sesquiterpene hydrocarbon | ϒ-elemene |
| 1,5-hexadiene | Oxalic acid | Ethyl cis-4-octaenoate | 11-hexadecanal | Styrene | | α-muurolene | (*Z*)-β-farnesene |
| Diethylbenzene | 2-methylpropanoic acid | Pentadecane | Ethyl 9-hexadecanoate | 3-hydroxybutan-2-one | | (*Z,Z*)-α-farnesene | 3-methylbutanoic acid |
| Methanol | 2-methylbutanoic acid | Cyclo-pentadecane | Benzyl octanoate | 3-methylbutyl-3-methylbutanoate | | a α-farnesene isomer; two | (*E*)-β-farnesene |
| n-pentanol | | β-Caryophyllene | (*Z*)-13-octadecenal | Hexan-1-ol | | Sesquiterpene hydrocarbons | Aromadendrene |
| 3-methoxy-s-butanol | Ethyl-2-propenylether | Hexadecane | Cyclodecene | (*E* + *Z*)-3hexen-1-ol | | (*E,E*)-α-farnesene | β-caryophyllene |
| 2-methyl-1-pentanol | Ethylisopropenylether | Isoamyl octanoate | Benzyl metacrylate | An ethylester | | δ-cadinene | A sesquiterpene hydrocarbon |
| 2-ethoxypropanol | Ethylamine | α-humulene | 6-dodecen-1-ol | Trans-linalool oxide furanoid | | ϒ-cadinene | heptadecan-2-one |
| 2-ethyl-3hexen-1-ol | Acetamide | Ethyl trans-4-decenoate | | Furfural; alkylbenzene | | A sesquiterpene hydrocarbon | nonadecan-one |
| Acetone | Ethyl isobutyrate | Heptadecane | | α-cubebene | | Ethylnicotinoate | α-bergamotene |
| Acetaldehyde | Ethyl valerate | (*Z*)-3-decenyl acetate | | δ-elemene | | Geraninol | pentadecan-2-one |
| Glycoladehyde | Ethyl butanoate | Heptadecene | | α-ylangene | | A sesquiterpene hydrocarbon | |
| Crotonaldehyde | Ethyl isovalerate | Germacrene D | | α-copaene | | Benzenemethanol | |
| n-pentanal | Ethyl propanoate | Benzyl acetate | | β-bourbonene | | benzene-ethanol | |
| 3-methylbutanal | 2-ethylbutanal | (*Z*)-3-decen-1-ol | | An ethyl ester | | Calcorene | |
| 2-hexenal | | 1-octen-3-yl butyrate | | β-cubebene | | 2,5-furandialdehyde | |
| n-hexenal | Ethyl formate | Benzyl butrate | | Benzaldehyde | | A sesquiterpene hydrocarbon | |
| n-heptanal | n-octanal | Nonadecane | | Linalool | | dodecan-1-ol | |

### 3. Improvement of Traditional Processing of Marula Fruits to Enhance Juice Yield, and the Flavour

The traditional processing techniques are inadequate to extract juice from the marula fruit. There is a need to improve the extraction process of marula juice to make the process cost-efficient. Generally, indigenous fruits are considered to be affordable food for the poor. Due to the insufficient information and knowledge about the potential health benefits and lack of standardised processing techniques, such fruits have become excluded from the main diets. These fruits are wasted due to the limited harvesting time, process control and storage conditions, which leads to variability in shelf-life and negative effects on the organoleptic properties, thereby reducing the nutritional quality of the product [43].

Through utilisation of knowledge gained in consultations with local marula wine producers and application of scientific fields (microbiology/biochemistry and biotechnology), the traditional marula winemaking is likely to shift from local (domestication) to medium industry. There is a clear understanding of the biological and biochemical processes that take place in marula winemaking, which involve selective harvesting of the fruits, juice extraction and fermentation. High juice yields require efficient marula juice processing conditions. In an effort to improve the extraction process, Hiwilepo-van Hal et al. [23] determined the optimum production conditions that favour high juice yield while considering the quality attributes of the juice, such as vitamin C content, polyphenols, antioxidant activity and the colour. Different parameters that were studied include temperature (25–60 °C), pectinase concentration (0.04–0.2%) and extraction time (5–65 min). The optimal extraction temperature for the vitamin C content and polyphenols as well as for antioxidant activity ranged between 40 and 60 °C. The vitamin C increased at a temperature above 40 °C and remained stable upon heating, even at higher temperatures. Polyphenols showed a proportional increase with an increase in temperature. For antioxidants, the radical scavenging capacity strongly depended on both vitamin C and polyphenols and the antioxidant increased remarkably at temperatures between 40 and 60 °C. The pectinase concentration in the range of 0.1–0.14% increased the yield of marula juice by 23% compared to non-pectinase extraction. It was also found that heating time had an effect on the lightness of the marula juice, while prolonged heating time changed the colour of the juice to darker yellow [23].

Fundira [22] focused on the influence of temperature, pectolytic enzymes and commercial yeast strains, which can enhance the typical flavour complex of the fruit in the fermented product, in an attempt to optimise the alcoholic fermentation (AF) of marula juice to achieve standardised quality of the fermented marula by-products. It was revealed that temperature directly influenced the AF by affecting the rate of yeast growth, and as a result, poor growth has led to a longer duration of fermentation. It also affected the biochemical reactions of the yeast species, which ultimately have an effect on the chemical composition and sensory quality of the wine [22]. The chemical and physical composition of the juice directly affects the rate and completeness of the fermentation as well as the resulting concentration of aroma and flavour constituents in the wine [22].

### 4. Microbiology of Marula Fruit Wine

Primarily, wine production is a microbiological process involving mainly yeast and bacteria. Microorganisms utilise sugars, amino acids and other nutritive compounds to produce the final product, wine. Wine production can occur through spontaneous fermentation or by using starter cultures.

#### 4.1. Spontaneous Fermentation

After the marula fruit juice is extracted, the popular practice is to mix the two separate juices (the concentrate and the diluted juice) and allow natural fermentation to take place. Thus, the fermentation process relied on the native yeasts present in the fruit juice. The process occurs as illustrated in Figure 3, and in the laboratory set-up, the conditions are

aseptic and controlled. Based on Phiri [27], the fermentation period of the two juices differs, with the more diluted juice taking longer to ferment (sluggish fermentation), and the alcohol content is also low (5–6.5% *v/v*). This low alcohol content was precise because of the low starting °Bx value, which represent total sugar content. The presence of non-*Saccharomyces* yeasts and bacterial species, which also utilise the sugars for their metabolic activities as well as nutritional components such as minerals, vitamins and nitrogenous substrates, may lead to insufficient support for the robust growth of the fermenting yeasts.

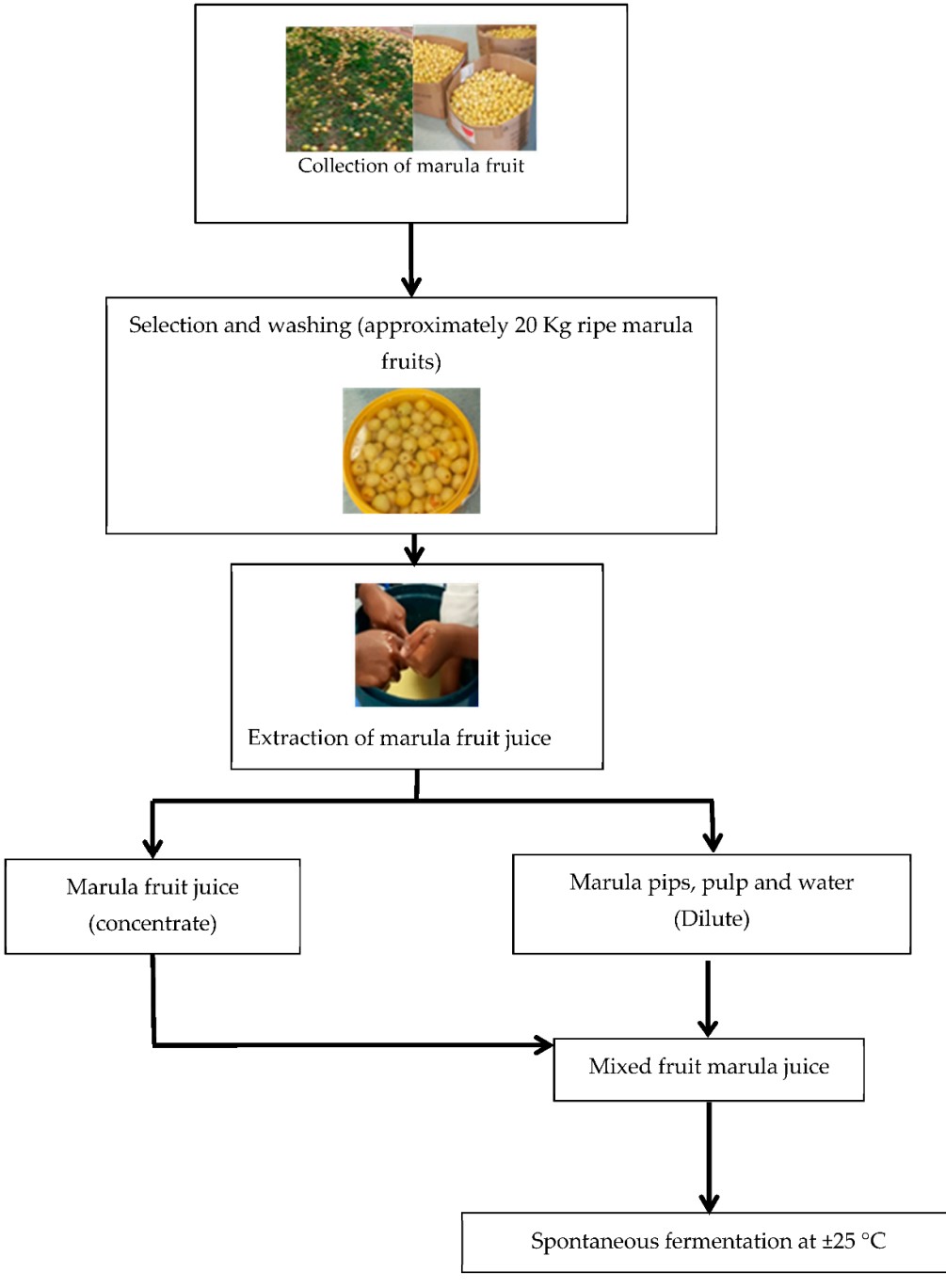

**Figure 3.** Production of marula wine from fruit.

Stuck fermentations are common in spontaneous fermentation due to nutrient deficiency, particularly nitrogen, which is required by the fermenting yeast. In some instances, marula fruit wine producers avoid stuck/sluggish/slow fermentations by using one container to produce different batches of the wine without cleaning it throughout the marula

fruits season. They believe that doing so will speed up the fermentation of the next batch because the microorganisms from the previous batch are already activated for fermentation. Some producers use fermented wine or half-fermented juice from a previous batch as a starter culture to fasten the fermentation. This process of adding a previously fermented product to initiate fermentation of the new product is known as back-slopping [44].

Other methods involve removing the peel from the fruit, and the juice is fermented together with the pulp on the nuts; alternatively, the fruit peel is cut open, and the whole fruit is fermented through spontaneous fermentation [45].

### 4.2. Microbial Evolution and Population Dynamics in Marula Wine during Spontaneous Fermentation

Fermentation of juice into wine is a complex microbial reaction with *Saccharomyces cerevisiae* as a candidate for AF and malolactic fermentation (MLF) involving lactic acid bacteria. The microorganisms on the skin of the fruit, which are introduced into the juice during the extraction process, are responsible for carrying out the fermentation process.

Wine fermentation is characterised by the evolution of a predominantly yeast population that is modulated by complex physical and metabolic interactions amongst various species. During spontaneous fermentation of marula juice, there is a succession of microorganisms involved in the process over a specified period of fermentation. On the first day of marula wine fermentation, the bacterial population is high, facilitating a decrease in pH from 4.1 to 3.88. At this stage, the dominant bacterial species isolated are Gram-positive rods, which are catalase positive and non-fermentative, belonging to *Lactobacillus* and *Leuconostoc*. As fermentation progresses, the bacterial count reduces due to the depletion of oxygen, while on the other hand, the fermenting yeast population increases during marula fruit wine fermentation [46].

Unlike in grape wine, the fermentation of marula juice is not fully characterised; however, recently, it was discovered that yeasts, lactic acid bacteria (LAB) and acetic acid bacteria (AAB) are involved from start to finish of fermentation. These microorganisms form complex microbiota that corporately acts together during fermentation. Maluleke [25] reported on the profile of the microbial community and the changes that occur during fermentation of the juice extracted from marula fruits obtained from different areas in the Limpopo Province, South Africa (the University of Limpopo, The Oak Village and Makhushane village). The findings confirmed the presence of bacterial and yeast species at different stages of fermentation. The common microorganisms in all three fermenting marula juices were bacteria species belonging to AAB, namely *Gluconobacter oxydans*, *Acetobacter pasteurianus*, and LAB, which according to [47] taxonomic re-classification, such microorganisms include *Levilactobacillus brevis* (formally known as *Lactobacillus brevis*), *Lactilactobacillus nagelii* (formally known as *Lactobacillus nagelii*), *Lentilactobacillus parabuchneri* (formally known as *Lactobacillus parabuchneri*), and *Lactiplantibacillus plantarum* (formally known as *Lactobacillus plantarum*) [25]. The yeast populations were *Hanseniaspora guilliermondi*, *Pichia guilliermondi*, *Saccharomyces cerevisiae*, *Rhodotorula mucilaginosa* and *Meyerozyma carribica* [25]. Given the common microbiota in all fermented samples, it was deduced that the intra-specific diversity remained low. At the early stages of fermentation *M. carribica*, *H. guilliermondi* and *L. brevis* and *L. plantarum* were dominant. As fermentation progresses, the dual concerted effort of the low pH and the continued accumulation of alcohol inhibited/or suppressed other microorganisms except for the *S. cerevisiae* and AAB, which dominated throughout the fermentation [25]. These observations were congruent with the findings of Phiri [27], who reported that the LAB, AAB and yeast remained present/or detectable for the duration of marula wine fermentation. The knowledge of microorganisms present in the fermenting marula juice and the chemical profile of the flavour compounds associated with such microorganisms can be beneficial for the development of starter cultures for use in both commercial rural households and industrial purposes.

The microbiota present on marula is also influenced by geographic location and climate. Okagbue and Siwela [48] identified the following microorganisms; *Aureobasidium pullans*,

*Geotrichum capitatum*, *Trichosporon brassicae*, *Rhodotorula mucilaginosa*, *Hansenula anomala* and *Hansenula jadini* from ripe marula fruits in Zimbabwe. None of these organisms except for *R. muciluginosa* were detected in both studies by Phiri [27] and Maluleke [25]. These microorganisms, based on their physiological characteristics, suggest that they are not playing a crucial role in traditional marula fruit wine fermentation. Surprisingly, there was no presence or detection of *Saccharomyces* yeasts [46]. Globally, there is an increasing interest in exploring non-*Saccharomyces* yeast and their potential usage in wine production [49]. The population dynamics in a multi-species consortium is strongly modulated by the response of each species to the presence of other species rather than external factors such as temperature [50]. It is an accepted notion that fermentation of either spontaneous or inoculated wines is ecologically complex, and apart from succession growth of non-*Saccharomyces* and *Saccharomyces* species, it also involves the successional development of strains within each species. The successional evolution of strains and species throughout the fermentation is largely determined by yeast susceptibility towards increasing concentration of ethanol and decreasing pH. Some non-*Saccharomyces* species exhibited tolerance towards increasing ethanol in wine [51]. Therefore, scientific knowledge of the key role players in marula fruit wine production would provide valuable information, which will allow researchers and winemakers to understand the chemical and sensory properties as they will be able to relate the microbiology and chemistry of the wine.

### 4.3. The Microbial Influence on Quality and Safety of Marula Wine

Processes involving microbiology and hygiene practices are key aspects of quality assurance in marula fruit handling and collection and fermentation. The fruit's colour changes from green to yellow at the onset of ripening [43]. The major challenge in assessing the marula fruit for processing is the difficulty in obtaining consistently ripe and undamaged fruit harvest. A common practice to date is using fruits that are having varying degrees of ripeness, and consequently, the product is astringent to the palate [23]. A stringent selection of ripe marula fruits should be the first step to ensure the high quality of marula wine. Some marula fruits, as they fall, the skin becomes damaged, making them susceptible to microbial contamination and insect infestation, which potentially cause loss of sensory (mainly taste), nutritional and other properties of the fruits.

Marula fruit and extracted juice are habitats of complex microbiota, which through species interactions during spontaneous fermentation such microorganisms contribute to the organoleptic properties of the wine. The emergence of starter cultures for use in winemaking could improve the organoleptic and unique sensory characteristics of the wine [52,53]. The knowledge of bacteria and yeast that are involved in the fermentation of marula juice is essential for the production of high and consistent quality wine [27]. During fermentation, yeasts produce flavour-active compounds, which are classified into six groups, including esters, fusel alcohols, ketones, various phenolics, aldehydes and fatty acids. The quality of the wine is affected if the concentrations of these volatile compounds in the final product exceed their taste thresholds [54,55]. The concentration level of volatile compounds depends on both yeast and bacteria, which impart the final wine character and aroma [50,56]. Although yeast predominantly produces high ethanol level and desirable aroma compounds, the lactic acid bacteria also play a crucial role in fermentation as they produce desirable acids, flavour compounds and peptides that inhibit the growth of undesirable microorganisms [57].

Lactic acid bacteria, particularly *Oenococcus oeni* produce acetic acid and acetoinic compounds ($C_4$ compounds) through citric acid metabolism, Figure 4 [11,16]. Acetoinic compounds are made up of diacetyl, acetoin and 2,3-butaediol, which can significantly influence wine aroma. Diacetyl (2,3-butanedione) is one of the most important flavour compounds produced during MLF and is known for its buttery or butterscotch aroma [11,16]. Lactic acid bacteria strains may also contribute negatively to wine quality by producing volatile phenols, biogenic amines and ethyl carbamate [11].

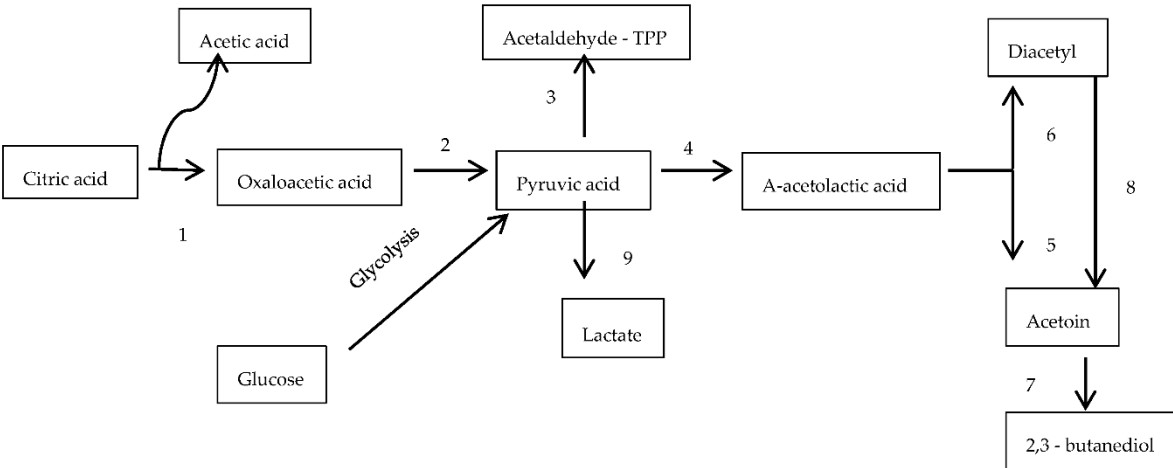

**Figure 4.** Simplified metabolism of citric acid by *O. oeni*. 1, citrate lyase; 2, oxaloacetate decarboxylase; 3, pyruvate decarboxylase; 4, α-acetolactate synthase; 5, α-acetolactate decarboxylase; 6, non-enzymatic oxidative decarboxylation; 7, acetoin reductase; 8, diacetyl reductase; 9, lactate dehydrogenase and TPP, thiamine pyrophosphate [11].

The conversion of citric acid by LAB leads to an increase in volatile acidity in wine (i.e., 1.2 molecules of acetic acid are produced from each molecule of citric acid). It is noteworthy to mention that the amount of acetic acid is minute and does not affect the quality of wine [16].

Esters represent the largest and one of the most important groups of aroma compounds produced during fermentation. They are associated with a diverse range of pleasant fruity flavours. Such attributes include banana and pineapple flavours. In addition, other flavour attributes such as citrus and floral can be characteristics of the presence of esters [58]. Esters are produced primarily by yeast(s). The concentrations of the esters are influenced by the fermentation process, mainly yeast, the composition of the must or juice and fermentation conditions. These aromatic compounds are formed by the esterification of alcohol (ethanol) and organic acid and acetyl coenzyme A (CoA), Figure 5 [16,59]. The rate of ester formation is dependent on the following factors: the concentration of the two co-substrates (i.e., the CoA component and alcohol) and the activity of the enzymes involved in their synthesis and hydrolysis [59]. Esters such as ethyl esters of branched fatty acids (i.e., ethyl 2-methylbutyrate, ethyl isobutyrate, ethyl isovalerate, ethyl 2,3 and 4-methylpentanoates) exert a concerted effect on wine aroma [16]. The microbiota of the grapes (including the marula fruit) is considered a pivotal factor that influences wine aroma and, eventually, consumer preference [16].

Although there are limited studies focusing on the influence of indigenous yeasts in marula wine, particularly the non-*Saccharomyces* yeast and their compatibility with *S. cerevisiae*. In grape wine, the non-*Saccharomyces* yeast species have shown the potential to produce wines with lower alcohol [60], mainly due to their non-fermentative nature and utilisation of the sugars prior to alcoholic fermentation. Again, the positive effect of non-*Saccharomyces* yeast depends on the concentration of the metabolites formed [61]. The marula fruit wine has unique sensory properties contributed by the microbial consortia occurring during fermentation. However, the secondary metabolite profiles are yet to be fully elucidated [62]. Spontaneous fermentation of marula fruit wine produces moderate concentration of volatile compounds such as acetaldehyde (6.16 mg/L), total ester (28.08–39.2 mg/L), total volatile acidity (201.48–654.4 mg/L) and higher alcohols (635.75–654.47 mg/L). These concentrations depend on physical parameters such as temperature and yeast strain type [53].

Organic acids are critical to the final flavour, colour and aromatic properties of the finished product. However, other acids such as acetic acid, propanoic acid, formic acid and butyric acid are undesirable in concentrations above their respective taste threshold as they impart off-flavour [63]. Acetic acid is the main acid constituting volatile acidity with its

threshold limit of 0.7–1.2 g/L; however, OIV (Organisation Internationale de la Vigne et du Vin) extended the limit to 2.1 g/L depending on the wine style [64]. The amount of volatile compounds produced by yeasts depends not only on the factors that regulate the quantity of CoA in the wine but is also influenced by the fermenting yeast, fermentation conditions, etc. Table 4 shows some odour-active compounds produced during marula winemaking by selected commercial starter cultures (VIN 7 and VIN13) and spontaneous fermentation (SPONTFERM) [53].

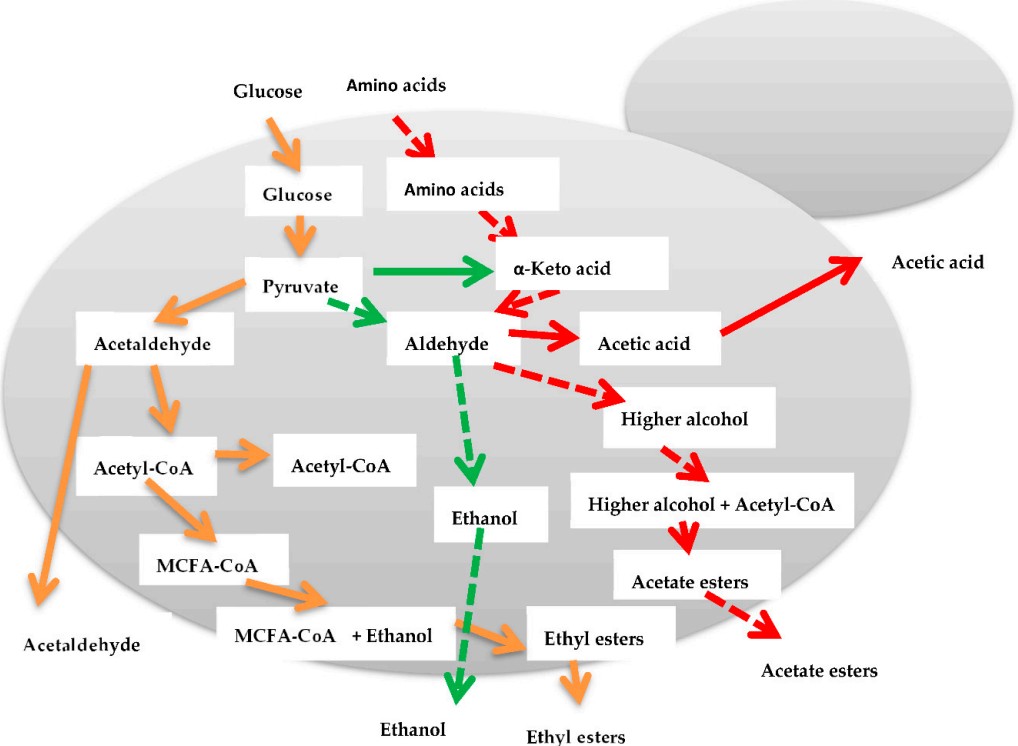

**Figure 5.** Formation of esters and other volatile compounds by *S. cerevisiae* through metabolism of glucose and amino acids.

**Table 4.** Volatile compounds produced in marula fruit wine by spontaneous and starter cultures fermentation [53].

| Compounds | SPONTFERM | VIN7 | VIN13 | Concentration in Wines | Sensory Threshold |
|---|---|---|---|---|---|
| **Higher alcohols (mg/L)** | | | | | |
| Propanol | 88.82 | 69.27 | 72.89 | 500 | 10–70 |
| Isobutanol | 36.43 | 35.28 | 42.18 | 10–150 | 40–50 |
| n-butanol | 1.58 | 1.15 | 2.01 | 0.5–10 | 150–200 |
| Isoamyl alcohol | 250.39 | 242.26 | 266.77 | 5–500 | 30 |
| 2-phenethylethanol | 76.72 | 84.41 | 81.26 | 5–200 | 10–15 |
| **Esters (µg/L)** | | | | | |
| Ethyl acetate | 28,120 | 20,220 | 19,560 | >150 | 10 |
| Isoamyl acetate | 460 | 220 | 260 | 100–3500 | 30 |
| Ethyl lactate | 3740 | 3260 | 3860 | - | - |
| 2-phenel acetate | 260 | 1410 | 200 | 18500 | 250 |
| Hexyl acetate | 0 | 130 | 0 | >5000 | 700 |
| **Fatty acids (mg/L)** | | | | | |
| Acetic acid | 646.80 | 348.05 | 430.69 | >1000 | 200–2100 |
| Isobutyric acid | 1.22 | 1.27 | 1.43 | Trace | 2.5–8 |
| Hexanoic acid | 1.43 | 1.41 | 1.95 | 0–40 | 2–8 |
| Octanoic acid | 1.90 | 2.02 | 1.97 | 0.4 | 0.5–10 |
| Decanoic acid | 1.51 | 2.34 | 1.20 | 0.5 | 1–10 |
| **Acetaldehyde (mg/L)** | 0 | 13.63 | 0 | 10–500 | 100–120 |

Microbial spoilage of the wine impacts negatively on the quality and hygienic status of the produced wine and, therefore, renders it unacceptable. The microorganisms associated with spoiled wine include (i) the yeast genera of *Brettanomyces*, *Candida*, *Hanseniaspora*, *Pichia* and *Zygosaccharomyces*; (ii) genera of the lactic acid bacteria such as *Lactobacillus*, *Leuconostoc*, *Pediococcus*; and (iii) the genera of acetic acid bacteria namely *Acetobacter* and *Gluconobacter* [65]. The spoiled wine has organoleptic faults, which include bitterness, off-flavours (i.e., mousiness, ester taint, phenolic, vinegary, buttery, geranium tone), turbidity, viscosity, sediment and film formation [65]. The presence and detection of such microorganisms necessitate the need for strict quality control measures to minimise their occurrence because their activity may lead to reduced product shelf-life and inconsistency in product quality. Consequently, this causes a significant loss for the industry [17,66].

Marula fruit wine studies by Phiri [27] explored the evolution of chemical compounds in the marula wines from different regions in the Limpopo Province from the onset of fermentation to the point where the wines were deemed unpalatable to taste. The study was able to detect and identify certain compounds during different fermentation stages and was able to correlate some of the compounds to the presence of microorganisms at a particular fermentation stage. The study detected nineteen volatiles present in marula wine after spontaneous fermentation. That included six alcohols (ethanol, 2-methyl-1-propanol, 1-propanol, 1-pentanol, hexanol and n-butanol), three aldehydes (acetaldehyde, isobutylaldehyde and formaldehyde), one ester (Isopentyl acetate) and nine organic acids (heptanoic acid, caproic acid, isocaproic acid, isovaleric acid, isobutyric acid, propionic acid, acetic acid and formic acid). The higher alcohols detected included 1-propanol, 1-pentanol, 2-methyl-propanol and hexanol [27]. There was a high concentration of acetate esters and aldehydes, including isopentyl acetate and isobutyraldehyde, in the early stage of fermentation only. Acetic and formic acid were also present in high concentrations throughout the fermentation process [27], and this could be attributed to the persistence of LAB and AAB (*Gluconobacter oxydans* and *Acetobacter* species), which remained present for the entire period of the fermentation. The specific organic acid composition of wine determines the specific pH of the wine, which in turn indirectly influences the perception of taste in wine [67]. The productions of some volatile compounds are the results of the dominant microorganisms producing compounds, which may also serve as a carbon source for other microorganisms [63]. The aromatic higher alcohols (also known as fusel alcohol) such as 2-phenylethanol, tryptophol and tyrosol are compounds derived from yeast metabolic activities during fermentation, which affect the aroma and flavour of the fermented beverages [68].

The role of commercial yeast starter cultures and their effect on the production of volatile compounds in wine and marula fruit wine was investigated. Wang et al. [69] reported different types of starter cultures with different effects on the flavour and quality of wine, and the findings revealed that different starter cultures adapt differently to wines of specific variety in specific wine-producing areas. When *S. cerevisiae* Zymaflore F15, *S. cerevisiae* 796, *S. cerevisiae* CEC01 and CECA were inoculated, the total acid was reduced, ranging between 6.2–6.9 g/L and 4.2–4.5 g/L acid in Cabernet Sauvignon wine [69]. Starter cultures in marula fruit wines also showed different aromatic profiles. For instance, *S cerevisiae* VIN13, VIN7, NT116, DY502, WE372, FC, WE14, N96, WE228 and DY10 produced different concentrations of acetaldehyde (2.75–21.72 mg/L) with the exception of strain FC which did not produce any acetaldehyde. The strains also produced different concentrations of higher alcohol, propanol (69.27–325.25 mg/L) in marula fruit wine. A higher total volatile acidy of 1508.44 mg/L was detected in marula fruit wine fermented with yeast strain DY502, and the lowest concentration was produced by strain WE372 (342.4 mg/L) [53]. The ability of a particular yeast starter culture to produce a specific aromatic compound was also influenced by the temperature under which wine fermentation is taking place [53].

The quality of wine is assessed by flavour as one of the major organoleptic characteristics [70]. In wine and beer beverages, flavour is the principal aspect of quality for

both producers and consumers [71]. Wine aroma is attributed to volatile and non-volatile compounds that combine to produce a specific effect. In grape wines, the complexity and variety of volatile compounds emanating from grapes interact with other non-volatile substances of the wine as precursors of wine's aroma known as primary aromas, which give the young wine its characteristics [72]. Most of these volatile compounds that are responsible for aroma are linked to sugars and form odourless glycosides. They are reverted into an aromatic form by hydrolysis. The chemical reactions amongst these compounds occur during fermentation to impart an aroma. Again, as wine ages, slower and gradual changes and developments in aroma take place [28]. Depending on the winemaking style, whether fermentation with marula peels contact or not, Table 3 exhibits various chemical compounds that, through chemical reactions and/or microbiological interaction with these compounds, could produce a specific type of wine with enhanced flavour.

In terms of safety, wine is considered a low beverage risk consumer product because of its inherent ability to suppress the growth of pathogenic microorganisms such as *Clostridium botulinum*, *Bacillus cereus* and *Clostridium perfringes*. Wine has high acidity (i.e., low pH of 3.1–3.9), polyphenols, alcohol (7–15% *v/v*) and low redox potential, and in some instances, sulphur dioxide is added. The above anti-microbial properties of the wine act synergistically to inhibit growth of microbial contaminants and ensure the high safety of the product [73]. LAB strains produce compounds with anti-microbial properties as well. *Lactiplantibacillus plantarum* HD1, which was isolated from Kimchi, produced three broad antifungal carboxylic acids: 3-hydroxy decanoic acid, 5-oxododecanoic acid and 3-hydroxy-5-dodecenoic acid [74]. The antifungal activities of these compounds ensure that the product remains free of fungal contaminants as these organisms could potentially release mycotoxins which pose a health risk, therefore making the wine unsafe for consumption.

## 5. The Sustainability and Economic Potential of Marula Wine

Indigenous alcoholic beverages are inexpensive to produce and are affordable for consumers of a wider range of income earners. Marula provides many benefits at the subsistence level, and there is an increasing interest in commercialisation and trading of its various products [6]. During marula season (January–March), hundreds of unemployed women begin to collect marula fruits to make traditional marula fruit wine. The rural communities, particularly in South Africa (Kwa-Zulu Natal, Mpumalanga and Limpopo provinces), have always relied on this natural resource, the marula tree, to alleviate poverty by generating income from the sales of fruits, marula fruit wine and kernels. However, marula fruit wine has always been the most popular product in the local commercial markets.

More than half of the women who participate in marula fruit wine making and trading have attained secondary education. Out of those with education, about 18% had a school leaving certificate and tertiary qualification [75]. Apart from the cultural ceremonies, the production of marula fruit wine is driven by a lack of employment, and therefore, this seasonal beverage is a way of generating income for households. According to Murye and Pelser [62], the continuing increase in the popularity of marula and its products on local and international markets led to the establishment of two marula processing plants in Swaziland. Due to high incidences of poverty, the initiatives had attracted a large number of Swazi women to scout the forests and fields around their homesteads in search of marula fruits and seeds to sell and earn income. Whilst this was seen as a way to alleviate poverty, the commercial harvesting of marula fruits poses a potential threat to the marula tree species as more seeds that would support the regeneration of new marula trees are removed through mass harvesting [62].

Marula fruits have numerous benefits for communities and provide subsistence to many households. The trading of its products and commercialisation thereof is slowly increasing [6]. In 2002, Shackleton [18] conducted a survey on the marula market involving fifty-one traders in the Bushbuckridge area of Mpumalanga. The survey intended to

discover and understand the commercialisation process, starting from the harvesting of raw material (fruit) to the marketing of marula wine. It became apparent that the sales of marula fruit wine provide an income to hundreds of households from the poorest within the communities. One important aspect of the marula business is that every local member can participate because there are no barriers to entry to trade since the marula fruit is abundant and freely available and accessible. The cost of producing marula fruit wine is minimal.

The pricing of marula products is highly variable as it is determined independently by marula winemakers in each household from one village (or province) to another. In previous years, marula wine sold in South African Rand, ZAR 5–7.50 per 2 L, and the sales depended on the exposure of the traders to the markets [2]. The selling price increased to ZAR 10.00 per 2 L [76], and nowadays, 2L of the wine sells for ZAR 21 and ZAR 25.00 in the Limpopo province.

## 6. Challenges, Opportunities and Future Research Recommendations

The economic feasibility and sustainability of marula wine production would require a high juice yield. Marula fruits are a valuable indigenous product of Africa and are available and easily obtainable at minimal cost for the production of marula wine. After the juice is extracted from the fruits, about 40–50% remains as pulp, which to date remains unutilised. In order to make the processing of marula fruits economically viable and cost-effective, the pulp should be converted to value-adding products.

The challenge in marula winemaking is the uncontrolled production environment, particularly in the communities that produce wines, which results in inconsistent quality of wine from one batch to the other. This inconsistency in the quality of wine hinders the scaling-up of the process. Standardisation of the marula wine production technique is necessary for reproducibility. Post-fermentation, several down-stream processing steps such as yeast removal, stabilisation and filtration are necessary to ensure high quality and longer shelf-life of the wine. Another factor that hampers large-scale production of marula fruit wine is the low yield of the juice. Exploration of strategies to enhance juice yield and preserve the juice would be beneficial prospects in the realisation of large-scale production of marula products such as fruit juice and wine.

The marula fruits are seasonal; therefore, preservation strategies for the fruits should be developed in order for the wine and other marula products to be available in the market throughout the year. There are currently few products made from marula juice on the market, with few attempts on artificial marula flavoured beverages being unsuccessful. A few studies have only focused on the chemistry of the juice to study the odour-active compounds [42], and no scientific information or little information is available on marula fruit wine. Furthermore, autochthonous microbiota of the marula, such as the yeasts and lactic acid bacteria, should be explored as potential starter cultures in the production of good marula fruit wine and other fermented non-alcoholic beverages.

Application of biotechnology in the winemaking process, which would include several aspects such as monitoring of the microbial population, the use of selected starter cultures and control of undesired microorganisms, can bring solutions to the marula fruit wine industry. Over the past years, the selected starter cultures of *Saccharomyces* yeast have been used widely at a large scale in the wine industry for inoculated wines in order to reduce the risk of spoilage and also for predictability of the process. The recognition of the value added to the wine by the non-*Saccharomyces* yeasts led to a shift towards using a mixed culture of *Saccharomyces* and non-*Saccharomyces* yeasts under controlled conditions. This approach was seen as a biotechnological and practical way towards improving wine complexity and enhancing the specific character of wine [77]. A similar approach may be adopted for use in marula fruit wine, whereby inoculation with selected mixed cultures of yeast can be used. By using the knowledge gained from spontaneous fermentation, Fundira et al. [53] highlighted the importance of autochthonous microorganisms on the marula fruit, and careful selection of such microorganisms could be useful for the devel-

opment of starter culture with desired traits or for the genetic strain improvement. The application of commercial wine starter culture derived from grapes, cellar, etc., has been reported by Fundira et al. [53] for use in marula wine. The authors demonstrated that all the wine yeasts investigated were able to complete the fermentation of marula fruit juice and the yeasts produced different quantities of flavour compounds.

To date, there is an information gap on the selected autochthonous or indigenous marula starter cultures, both yeasts (Saccharomyces and non-Saccharomyces) and LAB strains for use and inoculation of marula juice to carry out the AF and MLF. This offers an opportunity for the marula wine industry or marula wine communities to begin to explore starter culture applications in marula winemaking. According to [51], the isolation, careful selection and inoculation of these indigenous strains should be based on their individual performance (monoculture) or mixed culture performance and compatibility (mixed-culture) to avoid sluggish and stuck fermentation. For mixed starter culture, this would increase the microbial diversity with the effect of enhancing the character of the wine [51]. The criteria of selection and development of yeast starter culture fall under three categories, namely, (i) properties that affect the performance of the fermentation process: the yeast must ferment fast, vigorous and complete the fermentation of sugars to high ethanol concentration (>8% *v/v*); (ii) properties that determine wine quality and character: it is imperative that the selected yeast produce balanced quantities of flavour compounds without affecting the quality of wine; and (iii) properties associated with the commercial production of wine yeast: the yeast must be amenable to large-scale cultivation. Overall, the desired traits of yeast starter culture include improved fermentation performance, improved process efficiency, improved control of wine spoilage microorganisms, improved wine wholesomeness and improved wine sensory quality [51].

Lactic acid bacteria are also an important group of microorganisms in winemaking, which are present on the grapes in low numbers [78] and are also found on marula fruits [25]. Developing LAB strain starter cultures would require the selection of LAB strains based on an ecological study and the broad characterisation of useful technological and physiological features of the isolated strains in order to obtain the most suitable strain(s) for industry application. The criteria for selecting the LAB strain for winemaking rely on the survival of the strain in the wine environment and the ability to convert malic acid into lactic acid [61]. Maluleke [25] isolated four LAB species from the fermenting marula fruit wine, namely, *Levilactobacillus brevis*, *Lactilactobacillus nagelii*, *Lentilactobacillus parabuchneri* and *Lactiplantibacillus plantarum*. These species have the potential to be MLF starter culture or contributing to the aroma complexity of the wine. Equally, the same species might be detrimental to the quality of the wine by producing off-flavours. However, the contribution of these species to the wholesomeness of the wine is yet to be established. Virdis [61] reported the use of *L. plantarum* and *Oenococcus oeni* as mixed cultures for MLF. Interestingly, *L. plantarum* is homofermentative for hexoses and produces no volatile acidity through metabolism of sugars while having the potential to degrade biogenic amines [61]. Therefore, the selected wild yeast starter culture and LAB strain starter culture with high compatibility with each other could assist in controlling the fermentation process and equally contribute to the richness and complexity of the wine.

A comprehensive understanding of biological and biochemical reactions occurring during fermentation and their control mechanisms will make it easy to control the stages or steps in the process. There is an opportunity to use an integrated approach of microbiological and molecular techniques to assess the diversity and safety of the microorganisms (yeasts and bacteria) associated with spontaneous fermentation in food and beverages. The application of sequence-based molecular technologies might help to detect and identify the presence of indigenous strains that could potentially pose risks to human health [79,80].

## 7. Conclusions

Product development from the underutilised marula fruit will not only serve as wine and nutritional supplement but will most likely also open new markets. Improving

the technology for fermenting marula fruit wine could enhance the quality of the wine. Additionally, introducing heat treatment to extract the juice might increase juice yield and some flavour derivative compounds into the juice, which might be precursors for enhancing the overall aroma of the wine. Marula fruit can be turned into various products that span medicine and food and beverages. Modern practices demand consistent production under stringent control to ensure high product quality and safety. Inadequate training and skills in the application of starter cultures in small-scale production of marula fruit wine, particularly in the rural communities, under the currently existing conditions, present major challenges, which in turn such challenges offer the microbiologists and technologists exciting opportunities to engage communities and transfer knowledge and training to small scale producers. There is a scientific knowledge gap in the chemistry of aroma development in fruit and the microbial influence on aromatic compounds in the wine. Application of the available knowledge and biotechnological tools currently used in grape winemaking could vastly assist in understanding the critical stages in the winemaking of marula wine and improving wine quality.

**Author Contributions:** L.M.L. conceptualised the idea and laid out the structure of the manuscript. L.M.L. took the lead in writing the manuscript. M.A.L. contributed to writing part of the manuscript. K.L.M.M. critically reviewed the entire manuscript and assisted with the layout of the paper. All authors have read and agreed to the published version of the manuscript.

**Funding:** No funding was obtained.

**Institutional Review Board Statement:** Not applicable.

**Informed Consent Statement:** Not applicable.

**Data Availability Statement:** Not applicable.

**Acknowledgments:** We acknowledge the Department of Biochemistry, Microbiology and Biotechnology at the University of Limpopo for support.

**Conflicts of Interest:** The authors declare no conflict of interest.

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
