# Peer review of "Morula Tree: From Fruit to Wine through Spontaneous Fermentation and the Potential of Deriving Other Value-Added Products"

_processes, doi:10.3390/pr10091706_

Round 1
Reviewer 1 Report
The review article deals with the potential of the process for the production of wine through spontaneous fermentation and the potential of deriving other value-added products from fruits of Morula Tree. The article falls within the scope of the journal, and is relevant to the field of bioprocesses. Given the trend towards biorefineries where natural resources from various regions of the world are being used to the maximum, both economically and sustainably, the article is relevant. Although due to its structure it could be better in the journal “Sustainability” or “Fermentation” of this same publishing house (MDPI).
1.- The order of the pages is not correct, they must correct it
2.- Table 2 on page 7 shows that Marula (Sclerocarya birrea) can contain vitamin C concentrations ranging from 62 mg/100g to 400 mg/100g. The range is very wide compared to other fruits in the same table.
What is this about?
A seasonal issue or depending on the region where it is grown?
How much would this characteristic affect the yields of other components of the fruit to be exploited?
could this imply important variations in its processing?
3.-Page 6 line 526, Taking into account that it mentions that "The conversion of citric acid by LAB leads to an increase in volatile acidity in wine" and what is observed in table 2, where the range of variation of the concentration of vitamin c is very wide,
Would they be a potential problem for the quality of the wine?
4.- Page 6 line 537 and 537, which mean the “5 [xxx]. The” and line 540 “hydrolysis [xxx].”
5.- It is necessary to concentrate in a table the concentration ranges of alcohols, higher alcohols (also known as fusel alcohol) and organic acids that are produced from fermentation that give wine its characteristics.
6.- Page 8, line 603, change the word microbes for microorganisms
7.- Page 7, Figure 5, Improve the quality of the figure
8.- There is a lot of talk about fermentation and its implications, but the information on other value-added products is scarce and that conflicts with the name of the review
9.- Many bibliographic citations do not contain the DOI, in general the references should be reviewed and corrected
Reviewer 2 Report
This paper deals with the potential economic sustainability of the marul tree, S. birrea subsp. Caffra, its biochemical components of the fruits and juice. It also highlights the use of marula fruits for wine production in the African communities.
The authors in their review work emphasized the biochemical properties of marula juice, the flavour and aromatic compounds, improvement of traditional processing of marula fruits to enhance juice yield, and the flavors, the microbial evolution and population dynamics in marula wine during spontaneous fermentation and the economic potential of marula wine.
The manuscript follows correctly the template imposed by the magazine.
However, there are some mistakes in editing the figures. Also, the references are not numbering in the order that they are writing in the text.
The references are adequate.
Final recommendation:
The paper provides an interesting, original, actual and usefully research work. Yet, because of the above-mentioned observations and recommendations, this reviewer suggests minor revision of the manuscript.
